# SafeDec: Constrained Decoding for Safe Autoregressive Generalist Robot Policies

## Abstract

Recent advances in end-to-end, multi-task robot policies based on transformer models have demonstrated impressive generalization to real-world embodied AI tasks. Trained on vast datasets of simulated and real-world trajectories, these models map multimodal observations directly to action sequences for physical execution. Despite promising real-world capabilities, these models are still data-driven and, therefore, lack explicit notions of behavioral correctness. We address this gap by introducing **SafeDec**, a constrained decoding framework for autoregressive, transformer-based robot policies that enforces invariant safety specifications on candidate action trajectories. Task-specific safety rules are expressed as Signal Temporal Logic (STL) formulas and are enforced at inference time with minimal overhead. Our method ensures that generated actions provably satisfy STL specifications under assumed dynamics at runtime without retraining, while remaining agnostic of the underlying policy. We evaluate **SafeDec** on tasks from the CHORES benchmark for state-of-the-art generalist policies (e.g., SPOC, Flare, PoliFormer) across hundreds of procedurally generated environments and show that our decoding-time interventions are useful not only for filtering unsafe actions but also for conditional action generation. Videos are available at **constrained-robot-fms.github.io**.

## 1 Introduction

Recent advances in developing large transformer-based models for robotics have enabled general-purpose policies that map multi-modal inputs such as RGB images, natural language instructions, and proprioceptive inputs to action sequences (Hu et al., 2023). Shortest Path Oracle Clone (SPOC) (Ehsani et al., 2024), PoliFormer (Zeng et al., 2025), Flare (Hu et al., 2025) and OpenVLA (Kim et al., 2024) exhibit impressive generalization in navigation and manipulation tasks and serve as versatile robot controllers for real-world deployment contexts. However, these models are primarily data-driven and lack any explicit notion of safety. Although these models may implicitly exhibit safety-related behaviors depending on the patterns in their training data, there is no formal guarantee that models will consistently behave safely in all situations. This serves as a limiting factor for deploying these foundation models in the physical world where rule compliance and regulatory safety rule adherence are crucial.

Formal specifications have long been used to specify safety requirements for robotic deployments (Menghi et al., 2019; Farrell et al., 2018). Temporal logics (TL) (Pnueli, 1977) can capture safety constraints on robot behavior, such as "remain within the permitted region zones and avoid dangerous obstacles". Although TL has seen success in classical robotic planning for safety constraint satisfaction, its use for enforcing safety for large transformer-based robot policies remains limited. Additionally, retraining or fine-tuning these large pre-trained models to directly embed temporal logic specification is challenging (Kapoor et al., 2024). First, retraining models is a costly endeavor in terms of computational resources and data requirements. Moreover, due to the stochastic nature of these models, it is difficult to guarantee strict satisfaction of safety constraints through training alone. Hence, there is a pressing need for methods that can enforce safety specifications efficiently at inference time without disrupting the model's pre-trained behavior.

In the field of natural language processing, syntactic constraints have been successfully enforced by applying constrained decoding at inference time (Willard & Louf, 2023; Beurer-Kellner et al.,

2023; AI, 2023). These approaches typically mask out tokens that violate a syntactic constraint defined over token sequences. For example, regular expressions (regex) represent a widely used form of syntactic constraint, requiring that generated token sequences conform to predefined structural patterns (Willard & Louf, 2023; Beurer-Kellner et al., 2023). Inspired by this line of work, we extend the paradigm of constrained decoding to enforce safety constraints over action trajectories in dynamical systems and propose *safety specification aligned decoding (`SafeDec`)* for transformer based policies that ensures generated action sequences provably satisfy Signal Temporal Logic (STL) (Maler & Nickovic, 2004a) specifications under assumed dynamics. Our key insight is that decoding-time interventions can be used not just to filter unsafe actions, but to *condition the generation process itself* on specification satisfaction. This conditioning is critical because it steers the model toward generating safety specification satisfying actions rather than relying on post hoc rejection. `SafeDec` reduces risk of infeasible outputs while preserving the original action distribution of the model. To enforce such specifications, we leverage the formal semantics of STL to evaluate candidate actions at runtime and mask those that lead to future violations. Our method is agnostic to the underlying foundation model, requiring only two properties: (1) access to the decoding-layer logits during inference, and (2) access to an approximate dynamics model to predict future states. To efficiently evaluate STL specifications at inference time, we use a high-performance computational graph based library `STLCG++` (Kapoor et al., 2025c). In this work, we focus on safety enforcement for navigation policies in indoor environments, and leave extension to manipulation settings for future work. We also focus on STL specifications without liveness operators as they capture a large class of safety constraints in robot learning (He et al., 2024; Yun et al., 2025; Kapoor et al., 2025b; Zhao et al., 2024). To the best of our knowledge, this is the first work to enforce formal safety on transformer-based robotic policies at inference time using constrained decoding.

Our main contributions are as follows: First, we formalize the general problem of enforcing safety constraints during inference for autoregressive robot policies. In this work, we use STL to represent safety requirements because it is expressive over continuous states and provides quantitative robustness. Additionally, we focus on transformer-based autoregressive policies given their widespread adoption. Second, we propose an inference-time technique that reweights or masks candidate actions using STL satisfaction scores in (Section 3.2 and 3.3). Finally, we demonstrate the effectiveness of our method on state-of-the-art object navigation models without modifying model parameters (Section 4).

## 2 PRELIMINARIES

### 2.1 SIGNAL TEMPORAL LOGIC

Signal Temporal Logic (STL) is an expressive framework for defining properties and reasoning over continuous time real-valued (Maler & Nickovic, 2004b) . Formally, $(\mathbf{s}, t) \models \phi$ denotes that a signal $\mathbf{s}$ satisfies the STL formula $\phi$ at time $t$. An atomic predicate of an STL formula is represented by inequalities of the form $\mu(\mathbf{s}(t)) > 0$. The truth value of the predicate $\mu$ is equivalent to $\mu(\mathbf{s}(t)) > 0$. Note that with slight abuse of notation, $\mu$ represents both the predicate and a function of the trajectory $\mathbf{s}(t)$. Any STL formula consists of Boolean and temporal operations on these predicates, and the syntax of STL formulas is defined recursively as follows:

$$\phi := \mu \mid \neg\mu \mid \phi \wedge \psi \mid \phi \vee \psi \mid \mathbf{G}_{[a,b]} \, \psi \mid \mathbf{F}_{[a,b]} \, \psi \mid \phi \, \mathbf{U}_{[a,b]} \, \psi$$

where $\psi$ and $\phi$ are STL formulae, $\mathbf{G}$ denotes the globally operator, $\mathbf{F}$ the eventually operator, and $\mathbf{U}$ is the until operator. For example, $\mathbf{s} \models \mathbf{G}_{[a,b]}\psi$ specifies that $\psi$ must be in all times in the given interval, $t \in [a, b]$ of the signal $\mathbf{s}$. Similarly, the operator *until* in $\mathbf{s} \models \phi\mathbf{U}_{[a,b]}\psi$ defines that $\phi$ must be true until $\psi$ becomes true within a time interval $[a, b]$.

Given a signal $s_t$ representing a signal starting at time t, the Boolean semantics of satisfaction of $s_t \models \phi$ are defined inductively as follows:

$$s_t \models \mu \iff \mu(s(t)) > 0$$
$$s_t \models \neg\varphi \iff \neg(s_t \models \varphi)$$
$$s_t \models \varphi_1 \wedge \varphi_2 \iff (s_t \models \varphi_1) \wedge (s_t \models \varphi_2)$$
$$s_t \models \mathbf{F}_{[a,b]}(\varphi) \iff \exists t' \in [t+a, t+b] \text{ s.t. } s_{t'} \models \varphi$$
$$s_t \models \mathbf{G}_{[a,b]}(\varphi) \iff \forall t' \in [t+a, t+b] \text{ s.t. } s_{t'} \models \varphi$$

Apart from the Boolean semantics, quantitative semantics are defined for a signal to compute a real-valued metric indicating *robustness*, i.e., the strength of satisfaction or violation. For the sake of brevity, the definition of robustness is provided in Appendix A.1.

## 2.2 CONSTRAINED DECODING IN TRANSFORMERS

A large variety of autoregressive transformer-based models generate final outputs by producing a probability distribution over the model vocabulary at each timestep. This distribution is generated by performing a softmax operation over the model's last hidden layer. Then, through the process of *decoding*, tokens are selected to maximize the overall likelihood of an output sequence. In standard decoding, this maximization can be performed by either greedily selecting the most probable token at each step or by using a beam search to maintain multiple high-likelihood candidates. However, this often leads to degenerate output sequences that are repetitive (Holtzman et al., 2019). A common approach is to use sampling strategies like top $k$ (Fan et al., 2018), and nucleus sampling (Holtzman et al., 2019) that introduce stochasticity to encourage more diverse outputs. Constrained decoding (Hokamp & Liu, 2017) modifies this probabilistic selection by pruning invalid tokens to ensure that the generated sequences satisfy predefined constraints. These constraints are often syntactic, such as regular expressions, JSON formatting, or programming language grammars (Welleck et al., 2024). There is also recent work on enforcing *semantic* constraints that ensure coherence of the output or alignment with specific knowledge bases (Peyrard et al., 2024). Formally, constrained decoding can be seen as maximizing the probability of the output sequence subject to a constraint $\mathcal{C}$: $\arg\max_{y \in \mathcal{Y}_\mathcal{C}} P(y \mid x)$ where $\mathcal{Y}_\mathcal{C}$ is the set of sequences satisfying $\mathcal{C}$.

## 3 SPECIFICATION-GUIDED CONSTRAINED DECODING

In this section, we introduce a novel problem formulation for SafeDec in autoregressive transformer-based robot policies. First, we highlight the challenge in specification checking for these policies in contrast to traditional syntactical constraint checking adopted by LLMs, and our solution to remedy it. Then, we propose two novel inference-time techniques for specification aligned decoding: *Hard Constrained Decoding* (**HCD**) and *Robustness Constrained Decoding* (**RCD**).

## 3.1 PROBLEM STATEMENT

As highlighted in the background section, existing techniques in constrained decoding for language models enforce *syntactic constraints* defined over tokens, such as conforming to a context-free grammar or matching a regular expression. In these setups, constraint checking can be performed in the model's token space.

In contrast, transformer-based policies operate in a physical environment and constraints (captured via temporal logic) are defined over state variables. Since a large class of end-to-end policies solely propose action sequences, specification checking can only be performed as actions are executed and the environment is simulated forward. In this case, constraint checking cannot be done solely in the token space and requires environmental feedback or a dynamics stepping function. Hence, we leverage an approximate first order dynamics function to compute specification satisfaction of different action sequences proposed by these policies.

Consider a discrete dynamical system with states $x_t \in \mathbb{R}^n$ and actions $a_t \in \mathbb{A}$ at time step $t$. The system's dynamics are defined by $x_{t+1} = f(x_t, a_t)$ where $f : \mathbb{R}^n \times \mathbb{A} \to \mathbb{R}^n$ maps the current state $(x_t \in \mathbb{R}^n)$ and a discrete action $(a_t \in A)$ to the next state $x_{t+1} \in \mathbb{R}^n$. This system is controlled by

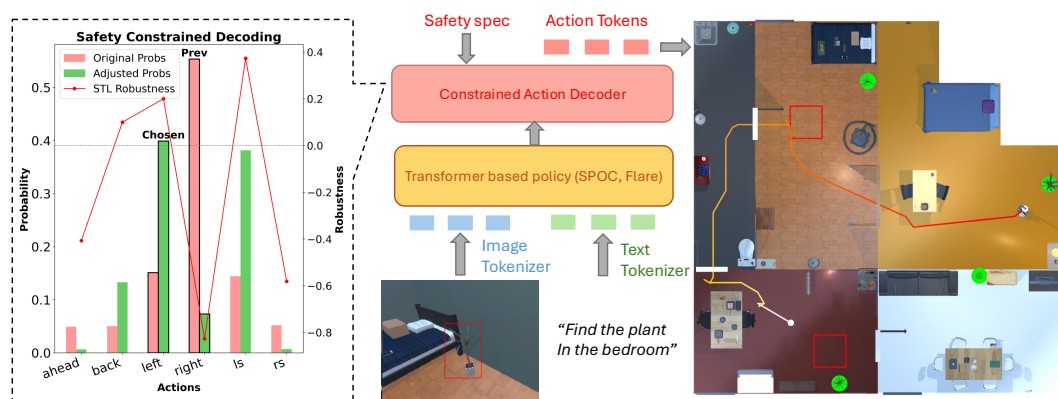

Figure 1: Overview of our specification aligned decoding framework. Given multimodal observations (RGB images and language goals), a pretrained transformer-based navigation policy (e.g., SPOC, Flare, PoliFormer) generates candidate actions. Our constrained decoder then filters or reweights these actions using robustness scores from a user-defined STL safety specification. **Left:** Original action probabilities (red) from the pretrained policy are modified by SafeDec using STL robustness: actions predicted to violate the safety specification receive reduced weights, while safer actions are boosted. The chosen safe action and adjusted probabilities (green) are highlighted. **Right:** An example navigation episode where the robot starts in the kitchen (white dot) and moves toward the bedroom to locate the target object (green marker middle-right), while avoiding user-defined hazardous regions (red zones) throughout the trajectory.

a policy that selects an action $a_t$ at each time step based on observations and task context such as user-provided natural-language instructions or goal waypoints.

In this work, we focus on robot policies that generate actions based on multi-modal inputs, including sensor observations (e.g., RGB, depth, LiDAR) and natural language instructions. Let $\mathcal{I}_t$ represent the aggregated input at timestep $t$. Given the history of encoded inputs up to time $t$, a transformer-based policy parameterized by $\theta$ predicts embeddings for the next $T-t$ actions:

$$\{\hat{e}_{a_{t+k}}\}_{k=1}^{T-t} = \text{Transformer}_\theta\big(\{e_{\mathcal{I}_\tau}\}_{\tau=0}^t\big).$$

Each predicted action embedding is decoded into an action $\hat{a}_{t+k} \in \mathbb{A}$, resulting in a predicted action sequence $\{\hat{a}_{t+1}, \ldots, \hat{a}_T\}$, where $T$ denotes the planning horizon.

Now, consider that the system is required to satisfy requirements encoded using an STL formula $\varphi$ defined over the state variables of the system. Formally, the goal is to ensure that the resulting trajectory satisfies the specification $\varphi$:

$$\{(x_0, \hat{a}_0), \ldots, (x_T, \hat{a}_T)\} \models \varphi$$

Most of the existing techniques for specification enforcement perform posthoc manipulation of proposed actions through filtering or rejecting action sequences that violate the specification $\varphi$. Although manipulation after sampling can ensure specification satisfaction, it can lead to distorting the model's learned distribution, producing low-likelihood output. This undermines the inductive biases learned during pretraining and leads to degenerate, brittle behaviors. A similar problem was highlighted when ensuring compliance with logical constraints for large language models (LLM) in Park et al. (2024). Additionally, these models decode actions sequentially, where each action $a_t$ is conditioned on previously generated tokens $a_{<t}$. Posthoc manipulation can disrupt this causal chain and lead to a mismatch between the model's internal hidden state and the executed sequence. Hence, we propose the following problem statement:

*How can we enforce temporal logic constraints during action generation in robot foundation models such that the output sequence (1) satisfies an STL specification $\varphi$, and (2) remains faithful to the model's autoregressive distribution $\pi(a_{1:T} \mid \mathcal{I}_{1:T})$?*

Let $\pi(a_{0:T})$ be the *unconstrained* action-sequence distribution produced by the transformer-based policy's decoder (e.g. the softmax over logits generated by the Transformer). We define the ideal constrained distribution over action sequences as:

$$Q_{\pi,\varphi}(a_{0:T}) = \frac{\pi(a_{0:T}) \cdot \mathbf{1}[(x_{1:T}, a_{0:T}) \models \varphi]}{\sum\limits_{a'_{0:T}} \pi(a'_{0:T}) \cdot \mathbf{1}[(x'_{1:T}, a'_{0:T}) \models \varphi]} \tag{1}$$

where $x_{1:T}$ denotes the state trajectory induced by the system dynamics under actions $a_{0:T}$ and $\mathbf{1}[\cdot]$ is the indicator function that returns 1 iff the trajectory-action pair satisfies the specification. Equation 1 is the exact Bayesian conditioning of $\pi$ on the event that the generated rollout satisfies $\varphi$. Hence, sampling from $Q_{\pi,\varphi}$ would give sequences that (i) inherit the original model's preferences encoded in $\pi$ and (ii) *guarantee* specification satisfaction.

In this work, we propose a technique to overcome the drawbacks of post-hoc safety enforcement methods (such as filtering) by leveraging constrained decoding techniques. Specifically, we propose `SafeDec` : A constrained decoding strategy that integrates STL specifications into the foundation model action selection process itself, ensuring satisfaction while remaining as close as possible to the base model distribution.

## 3.2 HARD CONSTRAINED DECODING

As highlighted in the background section, in the final layer, predictions are detokenized and a projection layer converts the embeddings into logits over the vocabulary space. These logits are further converted into a probability distribution using a softmax operation. In prior work, for structured output generation in LLMs, some invalid tokens are masked based on syntactical constraints or other criteria (Welleck et al., 2024; Park et al., 2024). This is done by setting their logit value as $-\infty$ before the softmax operation is applied. For HCD, we use a similar approach as constrained decoding literature (Welleck et al., 2024) and mask out predicted action tokens that violate our given STL specification $\varphi$ during sequential generation. Formally, to enforce the STL specification $\varphi$ during sequential generation, we adjust the logits at each timestep $t + k$ as follows:

Let $\mathbf{z}_{t+k}$ denote the logits at timestep $t + k$. For each action choice $i$ at timestep $t + k$, we define:

$$z_{t+k}^{(i)} = \begin{cases} -\infty, & \text{if } \hat{x}_{t+k}^{(i)} = f(x_{t+k-1}, \hat{a}_{t+k-1}^{(i)}) \text{ violates } \varphi \\ z_{t+k}^{(i)}, & \text{otherwise} \end{cases}$$

Here $t$ is the current decision step, $k$ is an index for the look-ahead step $t + k$ within a planning horizon of length $T$ ($k \in [1..T]$), $\hat{a}_{t+k}^{(i)}$ is the action mapping to the token i and $\hat{x}_{t+k}^{(i)}$ is the next state value upon taking this action. This next state is elicited using a simple dynamics model ($f$) as highlighted in the previous section. Adjusting logits in this fashion ensures that any invalid token with respect to the safety specification will have zero probability of being selected after applying the softmax function.

## 3.3 ROBUSTNESS CONSTRAINED DECODING

HCD ensures compliance but can lead to compromising task success, which can be undesirable. A similar tradeoff was observed by Liu et al. (2021) when probability space-steering preserved model fluency while reducing toxic continuations compared with hard-filtering strategies that inflated perplexity and eroded diversity. Hence, we propose an alternative approach, called RCD, where we leverage the quantitative semantics of STL specifications (robustness). Unlike HCD, which applies hard masking to completely remove unsafe actions, RCD softly guides the model toward safer actions by incorporating robustness scores that reflect the degree of satisfaction of $\varphi$. This is similar to the approach proposed in Liu et al. (2021) where the next-token distributions were re-weighted based on the utility scores provided by another language model. Our utility scores are quantified by the robustness function( $\rho(\langle x_0, x_1, \ldots, x_t \rangle, \varphi)$ ) that returns a real-valued score indicating how well a predicted state satisfies the specification. Positive robustness values denote specification satisfaction, while negative values capture the degree of violation. A formal definition of robustness in line with the STL quantitative semantics is provided in Appendix A.1.

First, we compute a robustness score for each candidate action: $r_{t+k}^{(i)} = \rho(\langle x_0, x_1, \ldots, x_{t+k-1}, \hat{x}_{t+k}^{(i)} \rangle, \varphi)$ where $\rho(\cdot, \varphi)$ is the STL robustness metric, and $\hat{x}_{t+k}^{(i)}$ is the predicted next state under action $\hat{a}_{t+k}^{(i)}$. This robustness score $r_{t+k}^{(i)}$ quantifies how well each candidate action satisfies the specification $\varphi$. These scores are then converted into weights using exponential scaling: $w_{t+k,i} = \exp(\alpha \cdot r_{t+k,i})$ where $\alpha$ is a temperature parameter that adjusts the sharpness of the bias. We use these weights to shift the original logits: $\tilde{z}_{t+k,i} = z_{t+k,i} + \beta \cdot w_{t+k,i}$ where $\beta$ is a hyperparameter that modulates the trade-off between specification adherence and the original task objective. Finally, we obtain the action distribution by applying softmax over the adjusted logits: $p_{t+k} = \text{softmax}(\tilde{\mathbf{z}}_{t+k})$

This approach allows for graded preferences that improve flexibility and robustness to dynamics approximation errors. By ensuring that all actions preserve a non-zero probability, the policy remains capable of recovering from errors arising from an imperfect dynamics model. Concretely, if the predicted successor $\hat{x}_{t+1}$ is off by $\epsilon$, an action that appeared marginally unsafe can be safe in the true system, and vice versa. Retaining a weighted down probability for this action gives the sampler a fall-back option whereas HCD would completely rule this action out due to 0 probability. Since we are shifting the probability mass for unsatisfying actions, it is possible that they are still chosen and lead to a violation. However, this is a tradeoff that we allow to achieve a given task objective. We note that this still ensures higher STL sastifaction than unconstrained actions.

## 4 EVALUATION

### 4.1 IMPLEMENTATIONAL DETAILS

We comprehensively evaluate our constrained decoding framework on procedurally generated AI2-THOR (Kolve et al., 2022) indoor scenes with diverse objects and layouts using three state-of-the-art (SOTA) generalist robot policies: Shortest Path Oracle Clone (SPOC) (Ehsani et al., 2024), PoliFormer (Zeng et al., 2025) and Flare (Hu et al., 2025). All three are large transformer-based embodied agents trained on extensive language-conditioned robot trajectory datasets. These models achieve strong zero-shot generalization for a vast variety of navigation tasks that span open vocabulary object-goal navigation ("find a mug"), room-to-room traversal ("visit all rooms"), waypoint-based navigation ("move three meters forward and stop near the red rug"), and attribute-conditioned variants ("locate the chair closest to the refrigerator in the kitchen"). These models also demonstrate reliable zero-shot transfer to real-world environments, achieving robust task satisfaction.

In addition, these models capture three different training paradigms for generalist robot policies. SPOC is trained purely with imitation learning from shortest-path rollouts. Poliformer employs a hybrid approach that combines reinforcement learning and imitation learning, enabling it to learn long-horizon structure while retaining expert priors. Flare adopts a large-scale pretraining plus fine-tuning on embodied navigation data in line with recent foundation model training paradigms. This diversity in training paradigms allows us to evaluate the applicability of `SafeDec` across different learning regimes.

In this work, we address safety specifications for robotics and, therefore, select those most relevant to real-world deployment. In particular, we focus on an important class of safety specifications called *invariants*, which are specifications that must be enforced at every reachable state of the system (e.g., "always avoid an unsafe region"). We enforce *geofencing* and *obstacle avoidance* by encoding them as invariant specifications in STL. Specifically, we generate random regions in the configuration space that the robot must either avoid (obstacle zones) or remain within (safe zones), and apply these constraints in real time during execution. The specifications used are: $\varphi_{\text{geofence}} = \mathbf{G}\left(\bigvee_{i=1}^{N}\left(x_i^{\text{L}} \leq x \leq x_i^{\text{U}} \wedge z_i^{\text{L}} \leq z \leq z_i^{\text{U}}\right)\right)$, $\varphi_{\text{avoid}} = \mathbf{G}\left(\bigwedge_{i=1}^{N} \neg\left(x_i^{\text{L}} \leq x \leq x_i^{\text{U}} \wedge z_i^{\text{L}} \leq z \leq z_i^{\text{U}}\right)\right)$. The size of the regions for $\varphi_{avoid}$ is $1\ m^2$. For $\varphi_{geofence}$, we randomly pick a subset of rooms in each house and use each chosen room's full bounds. These safety specifications are common for robot learning applications (He et al., 2024; Yun et al., 2025; Kapoor et al., 2025b; Zhao et al., 2024). We encode our test STL specifications using an efficient computational graph-based STL library called STLCG++ that can evaluate multiple state signals in parallel (Kapoor et al., 2025c). This ensures minimal inference overhead at

| Decoding | $\phi_{avoid}$: STL / SR (% ↑) | | | $\phi_{geofence}$: STL / SR (% ↑) | | |
|---|---|---|---|---|---|---|
| | **SPOC** | **Flare** | **PoliFormer** | **SPOC** | **Flare** | **PoliFormer** |
| Unconstrained | 72.0 / 82.5 | 75.5 / 82.0 | 77.0 / 82.5 | 78.0 / 81.5 | 68.0 / 81.0 | 73.0 / 81.5 |
| Filtering | 100.0 / 72.0 | 100.0 / 78.5 | 100.0 / 75.5 | 100.0 / 72.0 | 100.0 / 66.5 | 100.0 / 67.5 |
| HCD | 100.0 / 72.5 | 100.0 / 81.0 | 100.0 / 78.5 | 100.0 / 76.5 | 100.0 / 67.5 | 100.0 / 72.5 |
| RCD | 93.0 / 76.0 | 83.0 / 82.5 | 87.5 / 83.5 | 95.5 / 80.0 | 80.0 / 71.5 | 85.5 / 77.5 |

Table 1: Comparison by decoding technique across models (SPOC, FLARE, PoliFormer) for specifications $\phi_{avoid}$ and $\phi_{geofence}$. Each cell reports STL satisfaction / success rate (%). Higher is better (↑).

runtime ($10^{-5}$ s per timestep) , which is crucial for policy deployment. For our dynamics model, we assume a unicycle model, an approximate first-order dynamics abstraction widely used in the robotics literature for analysis and control (Cohen et al., 2024). This representation captures the essential kinematics of motion in the plane and is widely used because it is applicable for diverse robotic platforms.

## 4.2 EXPERIMENTAL SETUP

We compare our proposed techniques with (1) an unconstrained base model and (2) a base model with a filtering mechanism. The filtering mechanism picks a default action (turning left or right in place) upon predicted violation of the safety specification, similar to the Simplex architecture (Sha, 2001). Simplex architecture is a classic scheme in which a high-performance advanced controller is continuously monitored by a provably safe but less capable backup controller. Simplex based techniques have been used extensively for safety-critical robotics and are a widely accepted standard for runtime-safety comparisons. We evaluate performance using two main metrics: STL Satisfaction Rate (**STL St**), defined as the proportion of trajectories that satisfy the specified STL formula, and Task Success Rate (**SR**), which measures standard task success. The three main research questions we investigated in this paper:

1. **RQ1**: Do HCD and RCD provide higher STL satisfaction than the unconstrained baselines?
2. **RQ2**: Do HCD and RCD preserve task success rates comparable to the unconstrained baselines?
3. **RQ3**: Does RCD achieve better task success than HCD while maintaining high STL satisfaction?

## 4.3 RESULTS

Our results are highlighted in Table 1. We also visualize sample trajectories in Figure 2 for one scene and task. Unless stated otherwise, all numbers are averaged over 200 evaluation episodes.

**RQ1 – STL satisfaction.** Both HCD and RCD consistently improve STL satisfaction relative to the unconstrained baselines across all models. For $\phi_{avoid}$, unconstrained controllers achieve 72-77% satisfaction, while HCD raises this to 100% and RCD achieves 83–93%. For $\phi_{geofence}$, the gap is even larger: unconstrained models reach only 68–78%, whereas HCD attains perfect compliance (100 %) in all cases and RCD achieves 80–95%. We observe that the Simplex-style filtering baseline achieves similar STL-satisfaction rate as HCD, 97 % for $\phi_{avoid}$ and 100 % $\phi_{geofence}$. This parity is expected as both methods block any action predicted to violate the specification.

**RQ2 – Task completion.** Simplex-style filtering attains high satisfaction but sacrifices task success because the agent takes predefined safe actions. HCD shows similar behavior: although safety is maximized, success rates are consistently 5–10% lower than the baseline across models and specifications. However, as HCD factors in base model logits, it is able to achieve higher task satisfaction compared to Simplex-style filtering. In contrast, RCD preserves success rates much closer to the unconstrained level. For $\phi_{avoid}$, RCD achieves 82–85% success compared to 82–83% for the unconstrained controllers for Flare and PoliFormer. For $\phi_{geofence}$, it maintains 77–80% compared to 81–82% unconstrained for SPOC and PoliFormer. However, we note that RCD does not fully

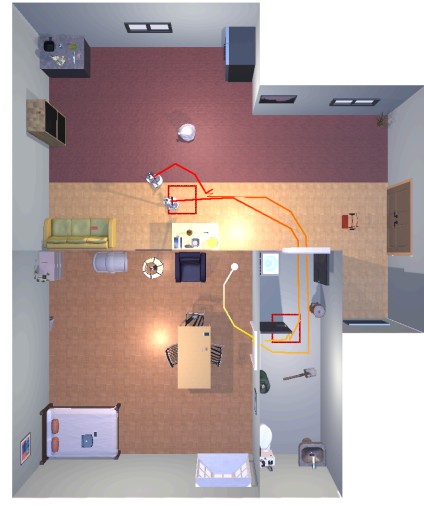
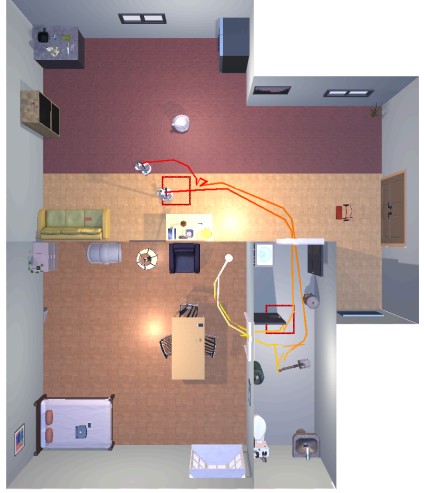

(a) Unconstrained v/s HCD                    (b) Unconstrained v/s RCD

Figure 2: Qualitative comparison of decoded trajectories for a sample scene. Each plot shows a top down view of an overlay of trajectories starting from the white dot under the instruction "find an alarm clock". The unconstrained model passes through two forbidden regions (red squares) on the way to the target object located on the table. In contrast, HCD (left) and RCD (right) modify the trajectories to respect STL safety specifications while still reaching the goal. More visualizations are available in the Appendix A.2.

recover success in every case: on SPOC with $\phi_{avoid}$ and Flare with $\phi_{geofence}$, task success remains several points below the unconstrained baseline. Nevertheless, RCD enforces safety while avoiding the large performance penalty observed with filtering.

**RQ3 – RCD vs. HCD.** While both HCD and RCD improve safety over the unconstrained baseline, they differ in how they balance constraint satisfaction with task success. HCD enforces strict STL satisfaction that results in frequent conservatism and lower successful task completion rates. In contrast, RCD's soft penalization leads to higher task success while still maintaining reasonable STL satisfaction. These results show that RCD achieves a better trade-off between safety and goal-directed behavior, especially in settings where occasional low-risk actions can lead to higher long-term rewards.

Overall, Our proposed techniques effectively enforce safety STL specifications during policy execution. HCD ensures full compliance, but occasionally sacrifices task success due to strict truncation. RCD strikes a balance, offering high satisfaction rates and robust performance. This highlights the feasibility of combining learning-based models with formal safety constraints.

## 4.4 ABLATION STUDIES

**Does inaccurate dynamics modeling substantially reduce STL satisfaction?** In this work, we assume a simple unicycle dynamics model due it's generalization capability for diverse robotic platforms. Although this represents a high-level abstraction of true dynamics, such modeling simplifications are standard in the formally assured robot safety literature (Cohen et al., 2024). However, both RCD and HCD depend on this assumption and inaccurate modeling can impact STL satisfaction. To evaluate the impact of inaccurate dynamics modeling, we conducted an ablation in which we inject gaussian perturbations into the dynamics (0.01 m per step translational noise i.e. 5% of nominal forward step, 1 ° per step rotational noise i.e. 3.3% of yaw step) for both HCD and RCD. Our results are visualized in Figure 3. Across all three base models, the drop in STL satisfaction rates from baseline to noisy dynamics is relatively small. We observe that `SafeDec` shows graceful degradation under significant per-step disturbances.

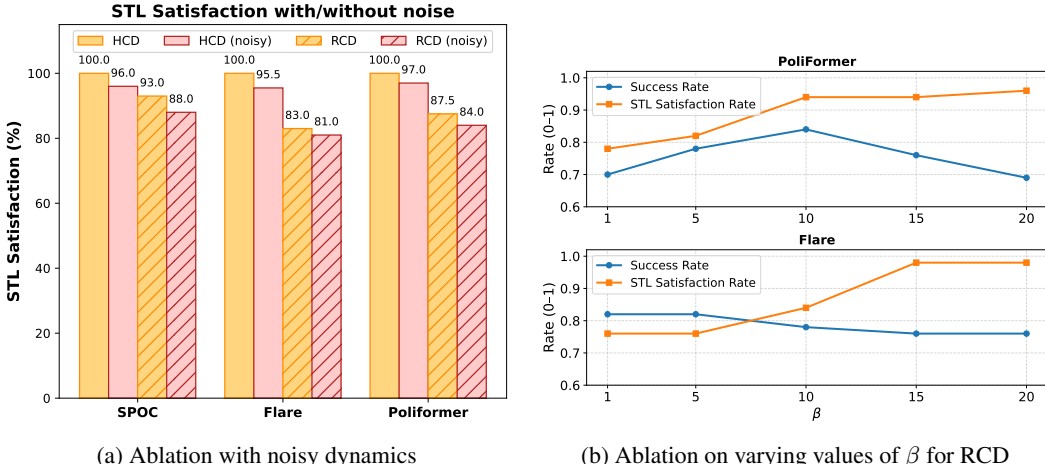

(a) Ablation with noisy dynamics      (b) Ablation on varying values of $\beta$ for RCD

Figure 3: **Ablation studies.** (a) STL satisfaction (%) for HCD and RCD under baseline vs noisy dynamics across base models. (b) Effect of $\beta$ on success rate and safety satisfaction.

**How do varying values of $\beta$ affect success rate and STL satisfaction in RCD?** As highlighted in section 3.3, RCD uses a hyperparameter $\beta$ that modulates the trade-off between specification adherence and the original task objective. The value of $\beta$ affects the relative weighting between robustness and base logits. To investigate the impact of $\beta$ on the success rate and STL satisfaction, we performed an ablation with varying values for $\beta$ for Flare and PoliFormer. We observe that as $\beta$ increases for PoliFormer, both STL satisfaction and success rate improve in tandem until $\beta = 10$, suggesting that moderate regularization can actually aid policy execution. Beyond this, STL satisfaction continues to improve but at the cost of lower success rates. For Flare, larger $\beta$ values improve STL satisfaction but reduce success rates. These results highlight that the influence of $\beta$ is model-dependent but in general demonstrate that `SafeDec` provides a tunable mechanism to balance safety and performance objectives.

## 5 RELATED WORK

Constraint satisfaction for robotics has been an active area of research that involves techniques such as control barrier functions (CBFs) (Ames et al., 2019), safe reinforcement learning (Gu et al., 2024), and temporal logic-based shielding approaches (Alshiekh et al., 2017). Recently, with the advent of vision language action models and their impressive generalizable capabilities for manipulation, navigation and other tasks, there are growing concerns about ensuring safety and correctness without retraining these large models. Although classical methods offer formal guarantees, they either require pretraining/fine-tuning stage interventions or designing a new classical controller for each safety specification, which can be restrictive. For example, SafeVLA (Zhang et al., 2025) fine-tunes pre-trained foundation models with task-specific safety costs, achieving strong performance in Safety-CHORES tasks. However, the safety specification is expected to be embedded in the training data and loss, meaning the model cannot generalize to new safety constraints at test time. In contrast, ASIMOV (Sermanet et al., 2025) explores rewriting dangerous instructions with better human-aligned alternatives to steer model behavior without modifying model parameters, but lacks trajectory-level formal guarantees. Our technique achieves a middle ground with the ability to adapt to novel specifications at test time without modifying model parameters while requiring minimal assumptions about the underlying model. The closest to our work is SELP (Wu et al., 2025) that proposes LTL-constrained decoding for language model-based plan generation. However, SELP is unsuitable for STL because its Boolean predicate-based LTL cannot encode numeric bounds (e.g. $||x-x_{goal}|| < 0.1m$) and does not possess quantitative semantics, which is crucial for ranking actions. These techniques are also tailored to high-level plans from LLMs, not to the per-step low-level actions generated online by policies.

## 6 LIMITATIONS AND FUTURE WORK

In this work, we introduce a constrained decoding framework for enforcing safety specifications for large transformer based robot policies. Our approach enables runtime adaptation to novel safety specifications without retraining. Through experiments across multiple simulated environments, we demonstrated that our method significantly improves STL satisfaction while maintaining high task success rates. Our approach makes two critical assumptions that can be a limiting factor. First, we assume access to specifications that are defined over the state space and that these specifications are generated by roboticists. Although this is a common situation for safety critical deployment contexts like aerial robotics (Aloor et al., 2023; Luckcuck et al., 2019), these specifications can be difficult to design and involve access to a localization module that can provide accurate state estimation. We plan to overcome this bottleneck by leveraging open-world safety specifications using recent work on embedding spaces-based logic (Kapoor et al., 2025a) and using large language models for generating high level specifications automatically (Li et al., 2025). Second, our approach also assumes access to an approximate dynamics model to evaluate the impact of actions on future trajectories. While a common assumption for provably safe robotics (Cohen et al., 2024), This can limit applicability of our approach. However, it is possible to mitigate this via learned dynamics models such as MoSim (Hao et al., 2025) or world models proposed in Zhou et al. (2025); Micheli et al. (2023) and we plan to explore this in future work.

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

# A    APPENDIX

## A.1    QUANTITATIVE SEMANTICS OF STL

Given a signal $s_t$ representing a signal starting at time t, the quantitative semantics of satisfaction of $s_t \models \phi$ are defined inductively as follows:

$$\rho(s_t, \mu_c) = \mu(x_t) - c$$
$$\rho(s_t, \neg\varphi) = -\rho(s_t, \varphi)$$
$$\rho(s_t, \varphi_1 \wedge \varphi_2) = \min(\rho(s_t, \varphi_1), \rho(s_t, \varphi_2))$$
$$\rho(s_t, \mathbf{F}_{[a,b]}(\varphi)) = \max_{t' \in [t+a, t+b]} \rho(s_{t'}, \varphi)$$
$$\rho(s_t, \mathbf{G}_{[a,b]}(\varphi)) = \min_{t' \in [t+a, t+b]} \rho(s_{t'}, \varphi)$$

## A.2    VISUALIZATIONS

To complement our quantitative results, we provide numerous trajectory plots from evaluations across a diverse set of procedurally generated indoor environments. Figure 4 illustrates representative top-down visualizations of trajectories induced by `SafeDec`. Each plot shows the agent's starting point and the resulting path under constrained decoding, with red circles marking target objects, green boxes denoting forbidden regions, and orange paths depicting the safe trajectories generated by our method. Together with our quantitative analysis, these qualitative results illustrate the performance of `SafeDec` across environments and tasks.

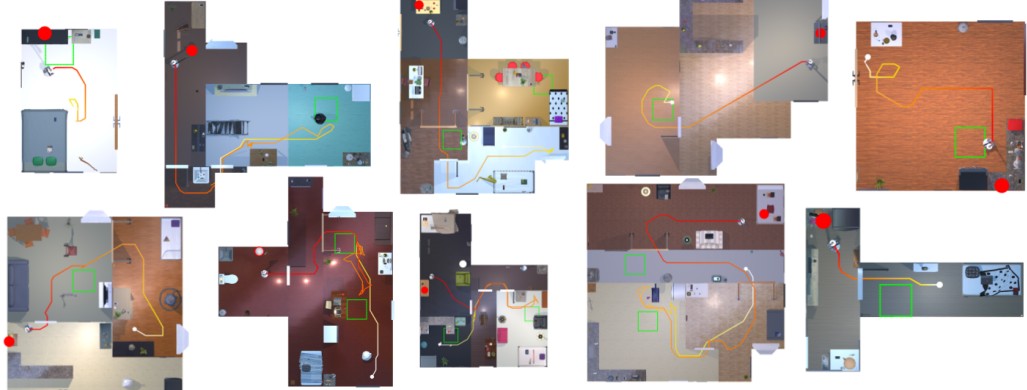

Figure 4: Top-down views across indoor environments with `SafeDec` induced trajectories Red circles mark target objects, green boxes are avoid regions, and orange paths show safe trajectories under constrained decoding.

