# OpenReview forum: "SafeDec: Constrained Decoding for Safe Autoregressive Generalist Robot Policies"
_ICLR.cc/2026/Conference — Submitted to ICLR 2026_

### Official Review · Reviewer_YNbh · 2025-10-16

**Soundness:** 2
**Presentation:** 1
**Contribution:** 2
**Rating:** 2
**Confidence:** 4

**Summary:**

This paper proposes SafeDec, a constrained decoding framework for robot policies. The robot policies are modelled as a pre-trained multi-modal causal transformer. The proposed framework controls the transformer’s output at inference time to ensure that robot behaviors satisfy safety requirements specified in Signal Temporal Logic (STL). Under this framework, the authors propose two techniques termed Hard Constrained Decoding (HCD) and Robustness Constrained Decoding (RCD). Assuming the system dynamics are available, HCD casts the logits for actions resulting in future failures to “-inf” to rule out unsafe behaviors, whereas RCD re-weights the logits for actions based on their contribution towards the robustness score as well as their original logits to balance the trade-off between safe behaviors and achieving tasks. The authors conduct experiments on AI2-THOR indoor scenes using three pretrained robot policies (SPOC, PoliFormer, and Flare) across two STL requirements. Compared with the unrestricted transformer baseline and filtering-based baseline, the proposed framework achieved a better trade-off between safety and goal-directed behavior. Ablation studies show that SafeDec can also handle inaccurate dynamics to some extent and show how different balancing coefficients for RCD lead to different performance.

**Strengths:**

1. The problem this paper aims to tackle is indeed valuable. Many robot policies now use transformer backbones, and one concern is the lack of safety guarantees. An efficient solution to enforce task constraints is of utmost importance.
2. The proposed framework is easy to understand and appears flexible to different robot systems. Since it doesn’t require the system to be differentiable, the proposed framework can handle broader types of safety constraints and system dynamics.

**Weaknesses:**

1. The idea is a bit trivial, given that the “constrained decoding” idea for Transformers has already been discussed in [1], and the experimental findings are not significant (the filtering baseline also achieves relatively high performance).
2. The experiment section lacks essential baselines (consider gradient-based methods [2,3], “beam-search” methods and sampling-based approaches such as CEM or CMA-ES [4], since the authors assume the system dynamics are available).
3. The proposed framework appears to work only on a discrete action space.
4. This framework is hard to generalize to STLs other than invariance properties.
5. The writing quality needs to be improved (misuse of citation types in L033-035; invalid citation in L084-085 and L301-303; wrong superscripts for “t’” used in L633-636; improper format for robustness score computation in L249-250 and L253-254, as the robustness score is on trajectories, not on states; L228-229 it should be $\hat{a}_{t+k-1}^{(i)}$) Some implementation details are missing (planning horizon T; the computation time, etc.)

References:
1. Willard, Brandon T., and Rémi Louf. "Efficient guided generation for large language models." arXiv preprint arXiv:2307.09702 (2023).
2. Leung, Karen, and Marco Pavone. "Semi-supervised trajectory-feedback controller synthesis for signal temporal logic specifications." 2022 American Control Conference (ACC). IEEE, 2022.
3. Zhong, Ziyuan, et al. "Guided conditional diffusion for controllable traffic simulation." arXiv preprint arXiv:2210.17366 (2022).
4. Kapoor, Parv, Anand Balakrishnan, and Jyotirmoy V. Deshmukh. "Model-based reinforcement learning from signal temporal logic specifications." arXiv preprint arXiv:2011.04950 (2020).

**Questions:**

1. I got confused for HCD introduced in L222-238. What is the k here, is it a hyperparameter defined for "look-ahead steps", or is it an index enumerated from 1 to T-t progressively? How do you get $x_{t+k-1}$ when you try to get $\hat{x}\_{t+k}$, I assume you only have $x_0,...x_t$ when you say "t is the current decision step"? Is the whole HCD process like, first from $x_t$, find possible actions $\hat{a}\_t$ that lead to STL-violation based on $\hat{x}\_{t+1}$, and then make these $z_t=-\inf$ so you never sample from them, and then sample from the rest buckets to get real $a_t$, and then use system dynamics to get $x_{t+1}$, then do the same thing to get real $a_{t+1}$ and then get $x_{t+2}$, ... till finding the final $x_{t+k}$? In this process, do you need to call the transformer forward pass multiple times? Do you need to update the transformer visual input? I guess the RCD procedure is somewhat similar. A pseudo-code algorithm or a simple example will be nice to illustrate the whole process.
2. In L213-215 the authors mentioned "... we propose SafeDec: A constrained decoding strategy that integrates STL specifications into the foundation model action selection process itself, ensuring satisfaction without distorting the model’s distribution." I am a bit confused why SafeDec doesn't distort the model's distribution. It seems like HCD will filter out some actions, and RCD will also change the action distribution by reweighting.
3. The reason for saying "Hard to generalize to STLs other than invariance properties" is that for the invariance properties without explicit time intervals, one just needs to treat them as state constraints per timestep, which is exactly how the current HCD and RCD handle constraints right now. For this type of constraint, one does not even need to use the concept of robustness score to reweight the logits. But for more complex STLs (like "Eventually (Always ...)"), it is no longer Markovian, and the agent needs to use extra bits to keep track of the history (as in some RL works[1,2], but only for two-temporal-layer STLs.) And for general STLs, one needs to first decompose them into reachability and invariant properties [3,4], then decides how to activate the associated subtasks as time progresses. The proposed framework does not seem to adapt to these settings.

References
1. Aksaray, Derya, et al. "Q-learning for robust satisfaction of signal temporal logic specifications." 2016 IEEE 55th Conference on Decision and Control (CDC). IEEE, 2016.
2. Venkataraman, Harish, Derya Aksaray, and Peter Seiler. "Tractable reinforcement learning of signal temporal logic objectives." Learning for dynamics and control. PMLR, 2020.
3. Kapoor, Parv, Eunsuk Kang, and Rômulo Meira-Góes. "Safe planning through incremental decomposition of signal temporal logic specifications." NASA Formal Methods Symposium. Cham: Springer Nature Switzerland, 2024.
4. Liu, Ruijia, et al. "Zero-Shot Trajectory Planning for Signal Temporal Logic Tasks." arXiv preprint arXiv:2501.13457 (2025).

---

> ### Author Response · Authors · 2025-11-22
> **Response to Reviewer YNbh**
>
> Thank you for the detailed feedback. We address weaknesses below and clarify several misunderstandings about the approach and contributions. We also indicate corresponding manuscript revisions.
>
> ---
>
> > **W1**: *“The idea is a bit trivial… already discussed in [1].”*
>
> **Response:**  We respectfully disagree that SafeDec is a trivial extension of prior constrained decoding. Prior work such as [1] focuses on symbolic/token-level constraints in text LLMs, while SafeDec is the first to show:
>
> - **STL robustness-based safety filtering** inside pretrained generalist robotic policies
> - Operating over state/action spaces, not text tokens
> - Online safety enforcement with zero retraining with provable safety guarantees modulo assumed dynamics
>
> Thus, SafeDec contributes a new online safety enforcement paradigm specific to *robotic policies*, with empirical validation on generalist navigation policies.
>
> [1] Willard, Brandon T., and Rémi Louf. "Efficient guided generation for large language models." arXiv preprint arXiv:2307.09702 (2023).
>
> ---
>
> >**W2**: *“The experiment section lacks essential baselines…”*
>
> **Response:** Thank you for the constructive feedback. We would like to clarify the scope mismatch. The suggested baselines operate in a classical control setting with full state observability, higher order dynamics models, and low-dimensional control spaces conditions under which gradient-based and MPC-style optimization methods are applicable. In contrast, SPOC, Flare, and PoliFormer are generalist, transformer-based policies that take **RGB observations and natural language commands** as input and maintain internal latent state, without providing access to full system state or dynamics. Because of this fundamental difference in interface and capabilities, the proposed optimization-based baselines are not fairly comparable to our decoder-level intervention setting.
>
>
>
>
> ---
>
> > **W3**: *“Works only on a discrete action space.”*
>
> **Response:**  Thank you for raising this issue. We acknowledge our technique only works for discrete action spaces in its current form. However, using a discrete action space is a common and effective assumption in modern embodied AI / VLA policies such as PACT [2], RT-2 [3], and policies with FAST tokenizer [4], all of which discretize continuous robot control into token-level actions for transformer-based sequence modeling.
>
> [2] Bonatti, Rogerio, et al. "Pact: Perception-action causal transformer for autoregressive robotics pre-training." 2023 IEEE/RSJ International Conference on Intelligent Robots and Systems (IROS). IEEE, 2023.
>
> [3] Zitkovich, Brianna, et al. "Rt-2: Vision-language-action models transfer web knowledge to robotic control." Conference on Robot Learning. PMLR, 2023.
>
> [4] Pertsch, Karl, et al. "Fast: Efficient action tokenization for vision-language-action models." arXiv preprint arXiv:2501.09747 (2025).
>
> ---
>
> > **W4**: *“This framework is hard to generalize to STLs other than invariance properties.”*
>
> **Response:**  Thanks for the constructive feedback, As part of the rebuttal, we have extended our specification diversity beyond pure invariants and added these specifications
> | STL Specification                         | Nesting Depth| Description |
> |:------------------------------------------|:-------:|:------------|
> | G(R1 ⇒ G(¬R2))                             |   1     | Avoid R2 permanently after entering R1 |
> | G(R0 ⇒ G(¬R0 ⇒ G(¬R0)))                    |   2     | Cannot return to R0 once exited |
> | G(¬R2) ∨ ((¬R2) U R1)                      |   0     | Avoid R2 always or until reaching R1 |
>
> These are not simple per-timestep invariants and require remembering past events and/or reach-avoid structure:
>
> The first two formulas are history-dependent safety properties: whether an action is safe at time t depends on whether the agent has previously entered/exited a region (non-Markovian from the state alone).
>
> The third formula uses an Until operator and that forces a temporal ordering for visiting these regions.
>
> ---
>
>
> > **W5**: *“The writing quality needs to be improved..”*
>
> **Response:**
> Thank you for the detailed feedback and engaging with our work.
> We corrected the noted items:
>
> - Fixed citation format and erroneous references (L34-35, L93-94, L 321-322)
> - Corrected superscripts and robustness notation (L34-35, L247 L267, L 270-272, L672-673)
>
>
> ---

---

> ### Author Response · Authors · 2025-11-22
> **Response to Q1-3 and Summary**
>
> ## Questions
>
>
> > **Q1**: *“…do you call the transformer forward pass multiple times?”*
>
> **Response:**  Thank you for requestting this clarification. k is an index enumerated from [1,T] where T is the length of the action sequence proposed by the model. Your assumption is accurate about propagation of state. Additionally, we call the transformer forward pass the same number of times the base model calls them to ensure one to one comparison. We will add a high level algorithm in the final version of the manuscript.
>
> ---
> > **Q2**: *“ I am a bit confused why SafeDec doesn't distort the model's distribution. It seems like HCD will filter out some actions, and RCD will also change the action distribution by reweighting.”*
>
> **Response:**  We appreciate this subtle point and have clarified the language in L231. We have rephrased our claims to emphasize that SafeDec aims to be as faithful as possible to the base model under the safety constraint, rather than leaving the base distribution unchanged.
>
> ---
> > **Q3**: *“For this type of constraint, one does not even need to use the concept of robustness score to reweight the logits. But for more complex STLs (like "Eventually (Always ...)"), it is no longer Markovian, and the agent needs to use extra bits to keep track of the history (as in some RL works[1,2], but only for two-temporal-layer STLs.) And for general STLs, one needs to first decompose them into reachability and invariant properties [3,4], then decides how to activate the associated subtasks as time progresses. The proposed framework does not seem to adapt to these settings.”*
>
> **Response:**  Thank you for this question. As stated in response to W4, we have added new specifications that are non markovian from state alone. Additionally, as pointed out by the reviewer JpmE, our approach focuses specifically on **safety-oriented STL specifications** without liveness operators and that it is still an important fragment to target. We have added the scope of specifications in the Introduction (L72-73). We explicitly state we are **not** addressing general liveness properties as those behaviors are better suited for the underlying base model that can operate with higher level inputs such as natural language commands. This allows our technique to complement the pretrained models with additional safety enforcement.
>
> ---
>
> ## Summary
> - Expanded STL experiments **beyond simple invariants**
> - Clarified model-distribution wording
> - Improved notation, citations, and formatting consistency
>
>
> ---

---

### Official Review · Reviewer_JpmE · 2025-10-20

**Soundness:** 3
**Presentation:** 3
**Contribution:** 3
**Rating:** 8
**Confidence:** 4

**Summary:**

The authors present an approach for satisfying invariant properties in plans generated by transformer models. Their approach uses an output filter that leverages the underlying distribution from the model, and mask predicted tokens that violate the invariant specification based on a hard constraint or robustness-based weighting. The latter provides a trade-off between satisfaction of the constraints and satisfaction of the original task, whereas the hard constraint formulation will sacrifice performance to satisfy the invariant property.

**Strengths:**

The authors present a sound, reasoned approach to safety using transformer models. The paper is well-written and easy to follow. The authors present the work well in the context of existing approaches, and it represents a nice first step to this type of online safety.

**Weaknesses:**

This paper overstates the application of STL. Invariant properties are a (useful) but small subset of STL. I think the abstract and paper need to be much clearer about this. It is a worthwhile step forward, but it is a very limited fragment of STL. After the abstract, invariants are not mentioned until section 4. Reading the paper, it sounds as though the authors will satisfy general STL specifications, which their proposed approach is not able to do. It is very important to be clear about this, as it represents a very concrete limitation of the approach.

It would be useful to see a comparison of the results against something like SafeVLA. That model requires fine-tuning with safety constraints, as the authors note. The existing comparators show the cost of satisfaction in terms of success rate. However, SafeVLA or a similar model could help the reader understand how close SafeDec comes to an optimal safe execution. That is, if SafeVLA outperforms SafeDec in terms of task satisfaction, then it helps a reader identify the tradeoff between fine-tuning for safety compared to run-time safety constraints.

**Questions:**

See weaknesses above.

---

> ### Author Response · Authors · 2025-11-22
> **Response to Reviewer JpmE**
>
> Thank you for the detailed and constructive feedback. We address each weakness  below and indicate the corresponding changes in the revised manuscript.
>
> ---
>
> ## Weaknesses
>
>
> > **W1**: “This paper overstates the application of STL. Invariant properties are a (useful) but small subset of STL. I think the abstract and paper need to be much clearer about this. It is a worthwhile step forward, but it is a very limited fragment of STL. After the abstract, invariants are not mentioned until section 4. Reading the paper, it sounds as though the authors will satisfy general STL specifications, which their proposed approach is not able to do.”*
>
> **Response:**  We appreciate this important clarification request and reviewer's acknowledgement of the work's contribution. As part of the rebuttal, we have extended our specification diversity beyond pure invariants and added these specifications
> | STL Specification                         | Nesting Depth| Description |
> |:------------------------------------------|:-------:|:------------|
> | G(R1 ⇒ G(¬R2))                             |   1     | Avoid R2 permanently after entering R1 |
> | G(R0 ⇒ G(¬R0 ⇒ G(¬R0)))                    |   2     | Cannot return to R0 once exited |
> | G(¬R2) ∨ ((¬R2) U R1)                      |   0     | Avoid R2 always or until reaching R1 |
>
> As pointed out by the reviewer, our approach focuses specifically on **safety-oriented STL specifications** without liveness operators. We have added the scope of specifications in the Introduction (L72-73).
>
>
>
>
>
> ---
>
>
> >**W2**:  *“It would be useful to see a comparison to something like SafeVLA… to help the reader understand how close SafeDec comes to an optimal safe execution.”*
>
> **Response:**  Thank you for the suggestion. We agree that SafeVLA offers a valuable reference point. We have now initiated comparisons and will add the results here as soon as they are available. We highlight that SafeDec is **training-free**, while SafeVLA **requires fine-tuning**.
>
> ---
>
>
> ## Summary
>
> - Added explicit scope statement: focus on runtime enforcement of STL safety specifications without liveness operators.
> - Expanded STL specification set to include conditional and temporally-dependent safety constraints.
> - Initiated SafeVLA comparison to evaluate training-free vs. fine-tuned safety performance.
>
>
> ---

---

### Official Review · Reviewer_qoXe · 2025-10-30

**Soundness:** 3
**Presentation:** 3
**Contribution:** 2
**Rating:** 4
**Confidence:** 4

**Summary:**

SafeDec introduces a constrained decoding framework that enforces formal safety specifications during inference for large transformer-based robot policies such as SPOC, Flare, and PoliFormer. Instead of retraining these models to internalize safety rules, SafeDec intervenes only at decoding time to ensure that generated action sequences comply with Signal Temporal Logic (STL) constraints.

**Strengths:**

- **Model-agnostic generality without retraining.** The framework operates at test time and is agnostic to the underlying policy; it is demonstrated on SPOC, Flare, and PoliFormer without any additional training or fine-tuning.
- **Clear empirical trade-offs between variants.** The simulation study contrasts HCD (strict safety satisfaction) and RCD (safety–performance trade-off), making variant-specific strengths and weaknesses evident.
- **Clear writing and easy to understand.**

**Weaknesses:**

- **Scope mismatch (navigation vs. manipulation).** While the introduction references both navigation and manipulation, all experiments are limited to navigation tasks in AI2-THOR, creating a gap between the paper’s claimed and demonstrated scope.
- **Problem framing conflated with solution design.** The claim of “introducing the novel problem of constrained decoding for transformer-based policies under STL specifications” blends the problem statement with specific solution choices (transformers and STL). The work would be more precise if it framed the broader problem as *online safety enforcement during inference* and justified these components as its methodological instantiation.
- **Limited evaluation.** Although the motivation centers on real-world safety, results are confined to a single simulated domain with two constraint types (geofencing and avoidance). This limits the generality of the conclusions.
- **No comparison with relevant baselines** (see questions)

**Questions:**

- **Ambiguity around “post-hoc manipulation.”** The term is used to critique existing methods but is not clearly defined—whether referring to post-training safety alignment or post-generation filtering at inference. Moreover, no empirical evidence from robotic settings supports the assertion that post-hoc interventions necessarily lead to “degenerate or brittle behaviors.”
- **Deviation from the base model’s distribution.** The method seeks to preserve the policy’s original action preferences while ensuring safety, yet both HCD (masking unsafe actions) and RCD (reweighting logits) modify the underlying probability distribution. This change conflicts with the stated goal of maintaining distributional faithfulness to the base model (lines 195--199).
- **Limited expressivity of tested constraints.** The evaluated constraints are simple invariants (“always avoid”). Demonstrating performance on temporally scoped constraints, such as “avoid room A between time 1--time 2” or “avoid A until visiting B”, would better motivate the need for STL’s representation.
- **Omission of baselines.** The paper only compares with bound-to-fail baselines and would be strengthened by comparisons against (a) multi-sample rollout selection (choose the trajectory satisfying safety), (b) corrective optimization of predicted actions to resolve violations, (c) described “posthoc manipulation” methods, and (d) recent guided-sampling approaches (e.g., DynaGuide).
- **Writing and presentation issues.** The reference style is occasionally inconsistent, and acronyms such as HCD and RCD should be expanded upon first mention in the abstract or introduction for clarity.
- **Missing experimental and implementation details.**  All rollouts were feasible under the specified constraints, including how infeasible cases were treated, and key implementation details such as action vocabulary, decoding horizon, sampling strategy, and mapping from STL predicates to action tokens. Including these would improve clarity and reproducibility.
- **Missing discussion on failure modes.**

---

> ### Author Response · Authors · 2025-11-22
> **Response to Reviewer qoXe**
>
> Thank you for the detailed and constructive feedback. We address each weakness below and indicate corresponding changes in the revised manuscript.
>
> ---
>
> ## Weaknesses
>
>
> > **W1**: *“While the introduction references both navigation and manipulation, all experiments are limited to navigation tasks in AI2-THOR, creating a gap between the paper’s claimed and demonstrated scope.”*
>
> **Response:**  Thank you for pointing this out. Our current evaluation focuses on navigation tasks for these policies because unsafe navigation (hazardous area entries, geofencing violations) is a common real-world safety concern for pretrained robot policies. We have revised the Introduction to clearly state that the experiments in this work are restricted to navigation tasks in AI2-THOR. (L70-72)
>
>
>
>
> ---
>
>
> >**W2**: *“The claim of ‘introducing the novel problem of constrained decoding for transformer-based policies under STL specifications’ blends the problem statement with specific solution choices (transformers and STL).”*
>
> **Response:**
> We agree the original phrasing could be more precise. We have rephrased the introduction to separate the problem from the design choices (L76-80)
>
>
> ---
>
>
> > **W3**: *“Results are confined to a single simulated domain with two constraint types (geofencing and avoidance), which limits the generality of the conclusions.”*
>
> **Response:**  Thank you for raising this point. To address the specification type diversity, we conducted additional experiments for complex specifications with temporal operator nesting, until operator and conditional clauses. The specifications and their results are provided in the general comment. For the simulation domain, our choice of AI2-THOR was motivated due to its realism and diversity. AI2-THOR houses are *procedurally generated* with diverse layouts and object placements, giving a wide range of navigation scenarios for SafeDec evaluation. We tested on 200 novel procedurally generated layouts. Additionally, policies such as SPOC, Poliformer trained in these environments have already shown zero-shot transfer to real-world navigation across diverse robots. We hypothesize that our technique will benefit from similar transfer capabilites observed by the base policies.

---

> ### Author Response · Authors · 2025-11-22
> **Response to Q1-4**
>
> ## Questions
>
> > **Q1**: *“The term is used to critique existing methods but is not clearly defined—whether referring to post-training safety alignment or post-generation filtering at inference. Moreover, no empirical evidence from robotic settings supports the assertion that post-hoc interventions necessarily lead to “degenerate or brittle behaviors.”*
>
> **Response:**  Thank you for requesting clarification. In this work, we use post-hoc manipulation to refer specifically to inference-time interventions that replace or alter an action after the model has already sampled it. Our filtering baseline performs such manipulation as it runs the base controller, detects predicted violations, and then replaces actions with a safe default. For the "degenerate behavior" claim, we add an explanation here. In certain cases, it is possible that post-hoc manipulation never allows the decoder to explore alternative viable modes of behavior. Autoregressive robot policies often represent multiple feasible strategies internally (e.g., different safe routes to a goal). But post-hoc filtering only evaluates the single sampled action. If this action is unsafe and must be rejected, the alternative mode becomes inaccessible can lead to safe behavior while making progress towards goals. As a result, post-hoc filtering can unintentionally suppress feasible safe behaviors that the model would have selected if safety constraints had been considered pre-sampling.
>
> ---
>
> > **Q2**: *“HCD and RCD modify the underlying probability distribution, which seems at odds with the goal of maintaining distributional faithfulness.”*
>
> **Response:**  We appreciate this subtle point and have clarified the language in L231. We have rephrased our claims to emphasize that SafeDec aims to be as faithful as possible to the base model under the safety constraint, rather than leaving the base distribution unchanged.
>
> ---
>
>
> >  **Q3**: *“The evaluated constraints are simple invariants (“always avoid”). Demonstrating performance on temporally scoped constraints, such as “avoid room A between time 1--time 2” or “avoid A until visiting B”, would better motivate the need for STL’s representation.”*
>
> **Response:**  We agree that showcasing more expressive STL formulas strengthens the case for using STL. In addition to the invariant geofencing and avoidance constraints, we have incorporated experiments with following specifications:
>
> | STL Specification                         | Nesting Depth| Description |
> |:------------------------------------------|:-------:|:------------|
> | G(R1 ⇒ G(¬R2))                             |   1     | Avoid R2 permanently after entering R1 |
> | G(R0 ⇒ G(¬R0 ⇒ G(¬R0)))                    |   2     | Cannot return to R0 once exited |
> | G(¬R2) ∨ ((¬R2) U R1)                      |   0     | Avoid R2 always or until reaching R1 |
>
> These specifications go beyond simple invariants and require reasoning over multi-step temporal structure (e.g., delayed consequences and conditional safety). We provide results in the general comment and will add them to the manuscript.
>
> ---
>
>
> >  **Q4**: *“Can the authors compare with multi-sample rollout selection, corrective optimization, post-hoc manipulation methods, and guided sampling approaches (e.g., DynaGuide)?”*
>
> **Response:**  Thank you for pointing us to these baselines. We already compare against the post-hoc manipulation methods (Filtering is an example of such methods). We are also running experiments for SafeVLA which is a baseline technique for constraint satisfaction of robot policies and will include them here.
>
> For corrective optimization, can the reviewer point us to the relevant literature so we can evaluate this technique? Additionally, we want to clarify why some of these are out of scope. The "multi sample rollout selection" depends on the base policy architecture and we tried to keep a one-to-one match with inference techniques of the base policy so our technique is easy to integrate. Upon reviewer's request, we tested this with Molmoact[1] and found that increasing the number of samples reduced task success relative to single-sample decoding, even before enforcing safety (likely due to distributional shift and hidden-state drift).
>
> DynaGuide is designed for diffusion policy based action heads and modifies sampling in continuous action spaces. Our models use discrete autoregressive action heads, making a direct comparison out of scope for this work.
>
> [1] Lee, Jason, et al. "Molmoact: Action reasoning models that can reason in space." arXiv preprint arXiv:2508.07917 (2025).
>
> ---

---

> > ### Author Response · Authors · 2025-11-22
> > **Response to Q5-7 and summary**
> >
> > > **Q5**: *“The reference style is occasionally inconsistent, and acronyms such as HCD and RCD should be expanded upon first mention in the abstract or introduction for clarity.”*
> >
> > **Response:**  Thank you for bringing this issue up, we have standardized the reference style, expanded acronyms at first mention (L145), and performed an additional proofreading pass.
> >
> > ---
> >
> > >  **Q6**: *“ All rollouts were feasible under the specified constraints, including how infeasible cases were treated, and key implementation details such as action vocabulary, decoding horizon, sampling strategy, and mapping from STL predicates to action tokens. Including these would improve clarity and reproducibility.”*
> >
> > **Response:**  Thank you for bringing up this point. We plan to release the code for the community to reproduce our results. We also answer the questions here for clarity.
> > - **Infeasibility case handling:** We are unsure what the reviewer means by infeasbility here. Can you please elucidate about this point?
> >
> > - **Action vocabulary:** For SPOC, Flare, and PoliFormer, actions are represented as a discrete set of primitives (e.g., move forward/backward, strafe left/right, rotate left/right, end, etc.). Each policy outputs a categorical distribution over this discrete action set.
> >
> > - **Horizon and sampling strategy:** We use the same sampling horizon as the underlying base policy and also the same sampling strategy (greedy or categorical distribution sampling). This is due to the challenges outlined for decoding in generalist policies compared to LLMs in Section 3.1.
> >
> > - **STL predicates to action mapping:** STL formulas are defined over *state variables* (e.g., agent pose \((x, z)\) in the floor plane). The approximate dynamics model is used to simulate the action outcomes and specifications are then evaluated over predicted state variables.
> > ---
> >
> > > **Q7**: *"Missing discussion on failure modes"*
> >
> >
> > **Response:** Thank you for your constructive feedback. The primary failure mode observed during experiments was early termination due to predicted safety violations. The action vocabulary has an "end" action usually used by the base policy to signify the end of an episode upon reaching target object. Since this action induces no changes in state, it can be the safest action when all other actions lead to violations and HCD/RCD rules them out. Hence, certain episodes ended too early without succesfully reaching target objects. Another failure mode is for specifications with implies operators. Since A ⇒ B = ¬A ∨ B,  it is possible to satisfy the  specification by never entering the region A. We observe this conservative behavior failure mode in certain episodes. We will add a discussion of these modes in the appendix.
> >
> > ---
> >
> > ## Summary
> >
> > - Clarified the scope of application to navigation tasks.
> > - Refined the  problem framing to separate the general inference-time safety objective from our specific instantiation using transformers and STL.
> > - Expanded the evaluation discussion and added more expressive STL formulas (temporally scoped and conditional constraints).
> > - Cleaned up writing and presentation, standardized references and expanded acronyms.

---

### Official Review · Reviewer_Pxk3 · 2025-11-02

**Soundness:** 2
**Presentation:** 2
**Contribution:** 2
**Rating:** 4
**Confidence:** 5

**Summary:**

The paper presents SafeDec, a constrained decoding framework that enforces formal safety specifications in Signal Temporal Logic (STL) within transformer-based robot navigation policies at inference time. The method is ​​an inference-time technique that reweights or masks candidate actions using STL satisfaction scores, i.e., decoding either through hard masking (HCD) or robust reweighting (RCD) of logits, ensuring that selected actions do not violate safety constraints under a simple dynamics model. SafeDec is evaluated on AI2THOR environments using three generalist navigation policies (SPOC, Flare, PoliFormer), demonstrating substantial improvements in STL satisfaction with limited impact on task success. The paper also compares performance when dynamics are noisy and the effect of weighting $\beta$ between robustness and base logits in RCD.

**Strengths:**

- SafeDec adapts constrained decoding from natural language processing to low-level action generation in robotic policies.
- The evaluation convincingly shows that SafeDec enforces simple invariants (avoidance, geofencing) with a modest performance loss across different state-of-the-art robot navigation policies.
- The paper demonstrates adaptation to small noise in dynamics and performance changes in the relative weighting of the safety–performance trade-off.

**Weaknesses:**

- The paper doesn’t compare performance to other similar methods mentioned (SafeVLA and SELP); instead, it is limited to simple filtering baselines.
- Implementation details are unclear, such as the setup of Simplex default actions and the decoding interface modification of the generalist policies. This affects the reproducibility.
- The paper has limited STL specification diversity, with only the Always operator (invariant) for “always avoid” and “stay within bounds” conditions being tested.
- The writing has stray citations (line 084, page 3, and lines 302-302, page 6 )and needs to be proofread once again. The explanation/caption of Figure 1 can be expanded to detail the architecture, the robot's initial and final positions, and the graph if the authors choose to place it at the beginning of the problem formulation section. Trajectory visualizations (Figure 2) have low contrast compared to the background and lack clear legends.

**Questions:**

1. How are actions represented in SPOC, Flare, and PoliFormer? Can you elaborate on how you access and manipulate the last-layer logits of these policies?
2. Can the paper include comparisons to other baselines, such as SafeVLA or SELP? Both explicitly address safety constraints in transformer-based robotics or planning.
3. Can the authors elaborate more on the mechanism of the default actions in the Simplex/Filtering baseline and how they were determined?
4. Can SafeDec experiments demonstrate performance with other STL specifications that use the eventually operator or until operator? Or larger compound specifications?
5. Have you tried experiments where the error in dynamic modeling was large due to unmodeled effects?

---

> ### Author Response · Authors · 2025-11-22
> **Response to reviewer Pxk3**
>
> Thank you for the detailed and constructive feedback. We address each weakness below and indicate corresponding changes in the revised manuscript.
>
> ---
>
> ## Weaknesses
>
> > **W1**: *“The paper doesn’t compare performance to other similar methods mentioned (SafeVLA and SELP); instead, it is limited to simple filtering baselines.”*
>
> **Response:**  In the interest of space, we provide the response for this weakness in Q2 since they have the same scope.
>
>
> ---
>
>
> > **W2**: *“Implementation details are unclear, such as the setup of Simplex default actions and the decoding interface modification of the generalist policies. This affects the reproducibility.”*
>
> **Response:** Thank you for raising this point. In the simplex baseline, we pick by randomly sampling from default actions that cannot immediately violate the specification (e.g., turning in place), which encourages the policy to explore alternative routes while still making progress toward the goal. For the decoding interface for each policy, we extract the logits for actions from the policy and then use the approximate dynamics to simulate action outcomes. Then we adjust probabilities based on predicted safety and sample from the updated distribution to get the final safe action.
>
> ---
>
>
> >**W3**:  *“The paper has limited STL specification diversity, with only the Always operator (invariant) for “always avoid” and “stay within bounds” conditions being tested.”*
>
> **Response:**   Thank you for pointing out the limited specification diversity. We conducted additional experiments for complex specifications with temporal operator nesting, untill operator and conditional clauses. The specifications are outlined in the general response and here:
>
>
>
> | STL Specification                         | Nesting Depth| Description |
> |:------------------------------------------|:-------:|:------------|
> | G(R1 ⇒ G(¬R2))                             |   1     | Avoid R2 permanently after entering R1 |
> | G(R0 ⇒ G(¬R0 ⇒ G(¬R0)))                    |   2     | Cannot return to R0 once exited |
> | G(¬R2) ∨ ((¬R2) U R1)                      |   0     | Avoid R2 always or until reaching R1 |
>
> These specifications go beyond simple invariants and require reasoning over multi-step temporal structure (e.g., temporal visit ordering and conditional safety).
>
> ---
>
>
> > **W4**: *“The writing has stray citations (line 084, page 3, and lines 302-302, page 6 )and needs to be proofread once again. The explanation/caption of Figure 1 can be expanded to detail the architecture, the robot's initial and final positions, and the graph if the authors choose to place it at the beginning of the problem formulation section. Trajectory visualizations (Figure 2) have low contrast compared to the background and lack clear legends.”*
>
> **Response:**  Thank you for the constructive feedback. We have made all requested changes as described below.
>
> **Changes in paper:**
> -  Corrected stray citations and performed a proofreading pass. (L93-94, L 321-322)
> - Expanded Figure 1 caption to describe:
>   - Overall architecture
>   - Robot’s initial and final pose
>   - Probability adjustment visualization
>
> ---

---

> > ### Author Response · Authors · 2025-11-22
> > **Response to Q1-4**
> >
> > ## Questions
> >
> >
> > > **Q1**: *“How are actions represented in SPOC, Flare, and PoliFormer? Can you elaborate on how you access and manipulate the last-layer logits of these policies?”*
> >
> > **Response:**  In all three policies (SPOC, Flare, and PoliFormer), actions are represented as a discrete set of low-level action primitives (e.g., move forward/backward, lateral strafing, rotation, end, and a few other discrete navigation actions exposed by each policy). At each timestep the policy outputs a categorical distribution over this shared action set. In our implementation we do not modify the architecture; instead, we extend the existing inference wrapper to expose the final action distribution. Concretely, the policy’s forward pass returns an actor_critic_output object whose distributions field already contains the per-action probabilities; we read out the corresponding logits.
> >
> > ---
> >
> > > **Q2**: *“Can the paper include comparisons to other baselines, such as SafeVLA or SELP? Both explicitly address safety constraints in transformer-based robotics or planning.”*
> >
> > **Response:**  Thank you for raising this point. We first discuss SELP beyond the differences brought up in (L481-485). While SELP is closely related in using formal logic for safety, it operates at a different abstraction level than SafeDec.
> >
> > - **Different control level:** SELP operates at the symbolic planning level and uses LLMs to generate high-level action sequences in natural language (e.g., “goto bedroom → goto kitchen”), whereas SafeDec enforces safety directly on the low-level navigation actions produced by pretrained embodied policies at every timestep.
> >
> > - **Offline specification compilation vs. online safety enforcement:** SELP requires translating the LTL specification into a Büchi automaton and enforcing constraints during plan generation, all of which is done offline. In contrast, SafeDec performs **inference-time** safety filtering online using STL robustness without any automaton construction or additional training.
> >
> > - **Cost and practicality:** SELP’s constrained decoding introduces significant planning-time overhead (6–8 seconds per plan in Table I)[0], whereas SafeDec modifies logits in milliseconds during action rollout, making it suitable for closed-loop execution.
> >
> >
> > In summary, SELP addresses safety before execution at the task-planning level using LLMs, while SafeDec enforces safety during execution inside transformer-based robot policies. Thus, the two approaches  are not directly comparable in experimental scope. Additionally, we have also reached out to the authors for their codebase since **they haven't open sourced their implementation** [1].
> >
> > [0] Wu, Yi, et al. "SELP: Generating safe and efficient task plans for robot agents with large language models." 2025 IEEE International Conference on Robotics and Automation (ICRA). IEEE, 2025.
> >
> > [1] https://github.com/lt-asset/selp/issues/4
> >
> >
> > **SafeVLA**: We are currently running experiments for SafeVLA and will add them here as soon as we have them. We emphasize that  our method is completely training free while SafeVLA involves fine tuning.
> >
> > ---
> >
> >
> > > **Q3**: *“Can the authors elaborate more on the mechanism of the default actions in the Simplex/Filtering baseline and how they were determined?”*
> >
> > **Response:**  In the simplex baseline, we label actions as “safe” if they cannot immediately violate the specification (e.g., turning in place), which encourages the policy to explore alternative routes while still making progress toward the goal. This action keeps a fair baseline with respect to safety while still letting the policy explore it's environment.
> >
> > ---
> >
> > > **Q4**: *“Can SafeDec experiments demonstrate performance with other STL specifications that use the eventually operator or until operator? Or larger compound specifications?”*
> >
> > **Response:**  As pointed out in response to W3, we have added three complex specifications. These specifications go beyond simple invariants and require reasoning over multi-step temporal structure (e.g., delayed consequences and conditional safety). We also wanted to highlight the scope of this work as enforcing safety specifications and not liveness specifications. This work focuses on **safely** deploying existing pretrained robot policies without disrupting their performance and modifying their parameters. The behavior that satisfies the liveness requirement is generated by the pretrained model (that can reason with high-level natural language commands and only RGB image inputs and explore its environment to satisfy them). Overriding the base model's behavior with additional liveness requirements could lead to suboptimal actions and undermine the base model's capabilities. Our approach adheres to the philosophy behind model-based safety methods, which deliberately decouple safety enforcement from liveness objectives. We have modified the manuscript and added the scope of specifications in the Introduction (L72-73).

---

> > > ### Author Response · Authors · 2025-11-22
> > > **Response to Q5 and Summary**
> > >
> > > >**Q5**:  *“Have you tried experiments where the error in dynamic modeling was large due to unmodeled effects?”*
> > >
> > >
> > >
> > >
> > > **Response:**  Thank you for raising this point. Yes, we evaluate SafeDec under modeling error by injecting Gaussian perturbations into the action-effect predictions used during decoding (Sec. 4.4). Since our assumed dynamics are already a high-level abstraction over the true AI2-THOR simulator (e.g., simplified kinematics without low-level physics), these perturbations simulate a range of unmodeled effects such as actuation bias, drift, and environmental disturbances. As highlighted in Figure 3(a), SafeDec continues to maintain high STL satisfaction under these perturbed dynamics.
> > >
> > > If there is a particular form of model mismatch you believe is critical, please let us know and we will run that variant as well and add the results here during the discussion period and in the final version.
> > >
> > > ---
> > >
> > > ## Summary
> > >
> > > - Clarified scope of safety baselines
> > > - Expanded STL specification coverage beyond invariants to specifications with conditional safety, untill operator and temporal nesting
> > > - Clarified implementation and decoding interfaces for SPOC, Flare, and PoliFormer.
> > > - Improved writing, fixed citations, and updated figure quality and captions.

---

### Author Response · Authors · 2025-11-22
**General comment**

We thank all reviewers for their thoughtful feedback and constructive suggestions. In this revision, we have run new experiments requested across all reviews, clarified implementation details, and improved presentation quality. Concretely, we:

- Added additional experiments to increase **STL specification diversity**, going beyond simple invariants to include history-dependent and temporal ordering specifications, as requested.
- Expanded and clarified implementation details of SafeDec, including how actions are represented in SPOC / Flare / PoliFormer, how we intercept and modify last-layer logits.
- Addressed writing and formatting issues, including fixing stray/invalid citations, standardizing reference styles, expanding the Figure 1 caption to better explain the setup.

We have uploaded an updated PDF; all modified text is highlighted in pink for easy tracking of changes. We also reply to individual questions in comments.

We also want to reiterate the key strengths of the paper that the reviewers highlighted:
- **Model-agnostic, inference-time safety:** *Model-agnostic generality without retraining. The framework operates at test time and is agnostic to the underlying policy; it is demonstrated on SPOC, Flare, and PoliFormer without any additional training or fine-tuning.* ~Reviewer qoXe
- **Modest safety-performance trade-off:** *The evaluation convincingly shows that SafeDec enforces simple invariants (avoidance, geofencing) with a modest performance loss across different state-of-the-art robot navigation policies.* ~Reviewer Pxk3
- **Flexible applicable mechanism:** *The proposed framework is easy to understand and appears flexible to different robot systems. Since it doesn’t require the system to be differentiable, the proposed framework can handle broader types of safety constraints and system dynamics.* ~Reviewer YNbh

# New specification results
We first highlight the specifications:
| STL Specification                         | Nesting Depth| Description |
|:------------------------------------------|:-------:|:------------|
| G(R1 ⇒ G(¬R2))                             |   1     | Avoid R2 permanently after entering R1 |
| G(R0 ⇒ G(¬R0 ⇒ G(¬R0)))                    |   2     | Cannot return to R0 once exited |
| G(¬R2) ∨ ((¬R2) U R1)                      |   0     | Avoid R2 always or until reaching R1 |

We added complex specifications with temporal operator nesting, untill operator and conditional clauses. These specifications, along with those in the original paper, span a broad class of invariance and conditional safety properties that are crucial for safely deploying pretrained policies.

Then we highlight the results for these specifications for Flare model as RCD performed the worst for Flare compared to SPOC and Poliformer for our original experiments. We observe similar results to our geofence and avoidance specifications for the decoding techniques lending further credibility to our original conclusions.

| Spec  | STL Formula                         | Decoder | STL satisfaction % |
|:-----:|:------------------------------------|:-------:|:------:|
| ϕ₁    | G(R1 ⇒ G(¬R2))                     | HCD     | 100    |
| ϕ₁    | G(R1 ⇒ G(¬R2))                     | RCD     | 93     |
| ϕ₁    | G(R1 ⇒ G(¬R2))                     | Vanilla | 74     |
| ϕ₂    | G(R0 ⇒ G(¬R0 ⇒ G(¬R0)))            | HCD     | 100    |
| ϕ₂    | G(R0 ⇒ G(¬R0 ⇒ G(¬R0)))            | RCD     | 96     |
| ϕ₂    | G(R0 ⇒ G(¬R0 ⇒ G(¬R0)))            | Vanilla | 82     |
| ϕ₃    | G(¬R2) ∨ ((¬R2) U R1)              | HCD     | 100    |
| ϕ₃    | G(¬R2) ∨ ((¬R2) U R1)              | RCD     | 89     |
| ϕ₃    | G(¬R2) ∨ ((¬R2) U R1)              | Vanilla | 72     |

# SafeVLA experiments
We appreciate reviewer feedback about comparing with SafeVLA as a way to understand how close SafeDec comes to an optimal safe execution while being completely training free.  We are currently running experiments for SafeVLA and will add them here as soon as we have them.

We hope these additions and clarifications address the main concerns raised by the reviewers and make the contributions and scope of SafeDec clearer.

---

### Author Response · Authors · 2025-11-28
**Follow-up message and additional SafeVLA experimental results**

I hope this message finds you well! As the discussion period draws to a close, we wanted to provide new results for SafeVLA experiments and check in to ensure we've addressed all your concerns and questions. We value your feedback and want to ensure we resolve any remaining issues so our work is as strong as possible.

Thank you again for your time and effort in reviewing our paper.

Upon reviewer PxK3 and jpmE request, we ran additional experiments to benchmark our technique against SafeVLA. SafeVLA proposes a safe RL based strategy that explicitly minimizes the cost of safety violations. The cost metric proposed in SafeVLA is a composite measure defined over multiple specifications such as maintaining distance from unsafe objects, avoiding fragile objects, blind spots, and corners, etc.

For our experiments, we focused on the fragile-object and dangerous-object specifications due to their close alignment in scope with our setting, and we constructed corresponding STL specifications for each. We report the cost metric proposed by SafeVLA for these sepecifications that captures violations of **both fragile-object and dangerous-object safety specifications**. The cost directly reflects the degree of violation of safety constraints across both object classes.

**Table: Performance Comparison (Lower Cost is Better, Higher Success is Better).**
Here, ISA corresponds to the SafeVLA policy, and FLARE is the baseline policy. The evaluation was performed on the SafetyChores benchmark proposed in SafeVLA for 200 episodes across various procedurally generated scenes.

| Method           | Cost ↓ | Success ↑ |
|------------------|--------|------------|
| ISA (SafeVLA)    | 0.205  | 0.865      |
| ISA + HCD        | 0.015  | 0.820      |
| ISA + RCD        | 0.060  | 0.860      |
| FLARE  | 0.192  | 0.820      |
| FLARE + HCD      | 0.115  | 0.840      |
| FLARE + RCD      | 0.155  | 0.790      |


**Key Takeaway.**
Both HCD and RCD substantially reduce safety violations relative to their respective vanilla baselines while largely preserving task success. In particular, integrating HCD with ISA (SafeVLA) achieves an order-of-magnitude reduction in safety cost (0.205 → 0.015), demonstrating effective enforcement of fragile and dangerous object specifications with minimal impact on task performance. We also find that RCD+ISA preserves task success close to just ISA while reducing safety violations. This experiment adds further evidence for our original hypothesis.

Additionally as pointed out by reviewer jpME:
>"However, SafeVLA or a similar model could help the reader understand how close SafeDec comes to an optimal safe execution. That is, if SafeVLA outperforms SafeDec in terms of task satisfaction, then it helps a reader identify the tradeoff between fine-tuning for safety compared to run-time safety constraints."

Our results show that SafeDec, despite being a training-free method, preserves task satisfaction close to the safeRL baseline (ISA) for our specifications. Additionally, combining SafeDec with the ISA policy leads to further safety cost reduction, highlighting its complementary benefits to data-driven safe learning.

---

### Meta-Review · Area_Chair_zPgw · 2026-01-07

**Summary:**

The paper introduces SafeDec, an inference-time framework for enforcing safety constraints in autoregressive robot policies. It operates by intercepting the logits of the policy during decoding and applying either a hard mask or a soft reweighting, verified against a simplified dynamics model. The review process was mixed. While reviewers appreciated the practical utility of a training-free safety filter and the clear empirical trade-offs shown in simulation, significant concerns regarding incremental novelty and scope ultimately weighed against acceptance.

**Reviewer Concerns:**

- Limited expressivity: Reviewers (Pxk3, qoXe) noted that the initial evaluation focused only on simple invariant constraints, failing to demonstrate the necessity of STL for complex, temporally dependent safety rules.

- Missing baselines: Reviewers requested comparisons against state-of-the-art safety methods, particularly fine-tuning approaches like SafeVLA, to properly contextualize the trade-offs of a training-free approach.

- Novelty: A critique (Reviewer YNbh) was that the method is a straightforward adaptation of existing constrained decoding techniques from NLP to discrete robotic actions, lacking substantial algorithmic innovation.

- Generality: The evaluation was restricted to navigation tasks within a single simulator (AI2-THOR) using discrete action spaces, which limits the validity of claims regarding "generalist" robot policies that typically require continuous control or manipulation capabilities.



The authors effectively resolved concerns regarding expressivity and baselines during the rebuttal. However, the critical concerns regarding novelty and scope remain outstanding.

**Reviewer Scores:**

The reviewers’ scores range from 2 to 6, demonstrating a skeptical consensus that is unlikely to improve with further discussion.The assessment appears static due to the unresolved conceptual and theoretical concerns.

---

### Decision · Program_Chairs · 2026-01-26

Reject